# Tissue block staining and domestic adhesive tape yield qualified integral sections of adult mouse orbits and eyeballs

Zhongmin Li[1]*, Martin Ungerer[2], Julia Faßbender[1], Clara Wenhart[1], Hans-Peter Holthoff[1], Goetz Muench[1]

**1** Advancecor GmbH, Martinsried, Germany, **2** ISAR Bioscience Institute, Planegg, Germany

* li@advancecor.com

**Data Availability Statement:** All relevant data are within the manuscript and its Supporting information files.

## Abstract

The standard histological processing procedure, which produces excellent staining of sections for most tissues, fails to yield satisfactory results in adult mouse orbits or eyeballs. Here, we show that a protocol using tissue block staining and domestic adhesive tapes resulted in qualified integral serial cryo-sections of whole orbits or eyeballs, and the fine structures were well preserved. The histological processing protocol comprises paraformaldehyde fixation, ethylenediaminetetraacetic acid decalcification, tissue block staining with hematoxylin and eosin, embedding, adhesive tape aided sectioning, and water-soluble mounting. This protocol was proved to be the best in comparison with seven other related existing histological traditional or non-traditional processing methods, according to the staining slice quality. We observed a hundred percent success rate in sectioning, collection, and mounting with this method. The reproducibility tested on qualified section success rates and slice quality scores confirmed that the technique is reliable. The feasibility of the method to detect target molecules in orbits was verified by successful trial tests on block immunostaining and adhesive tape-aided sectioning. Application of this protocol in joints, brains, and so on,—the challenging integral sectioning tissues, also generated high-quality histological staining sections.

## 1. Introduction

Recent work in visual neuroscience research has concluded that some degenerative eye diseases follow topographically specific retinal ganglion cell death across the retina [1–10] and that the anatomical orientation of the retinae with respect to the orbits is important [11–32]. These studies require visualization of serial whole integrated sections of orbits or eyeballs to determine morphological changes that respond to treatments or genetic modifications.

Whole integrated sectioning of orbits or eyeballs with either traditional paraffin or frozen OCT (optimal cutting temperature) embedding is rather difficult in postnatal mice older than 14 days, especially for relatively thin sections, even with an utmost attempt at precision and perfection [33]. The difficulty in preparing whole integral sections of the orbits or eyeballs

**Funding:** Support was provided by Advancecor GmbH of Germany, in the form of authors' (ZL JF CW HH GM) salaries and/or research materials. The funder had no role in study design, data collection and analysis, decision to publish, or preparation of the manuscript. The specific roles of these authors are articulated in the 'author contributions' section.

**Competing interests:** The authors have read the journal's policy and have the following competing interests: Julia Faßbender, Clara Wenhart, Hans-Peter Holthoff, and Goetz Muench are paid employees of Advancecor GmbH of Germany. There are no patents, products in development or marketed products associated with this research to declare. This does not alter our adherence to PLOS ONE policies on sharing data and materials.

arises chiefly because of tremendous differential shrinkage between orbital contents at different temperatures during cutting [34]. These contents of extremely different densities include intraorbital glands, lenses, vitreous humor, the high dense sclera, and the orbital walls of both mineralized and fibril bone tissues [34].

In order to improve current restrictions to secure the whole integral sections of orbital or eyeball, isolation [35–38] of separate tissues or enucleation [39] was performed to study local morphological changes. The technique using a softening lens by chemicals was also used for the sectioning of the eyeball [40]. However, these often lead to a high risk of missing or damaging some parts of tissue and losing orientation of the tissue during histological processing. A comparison between the different treatments is sometimes not valid without unified orientation.

To obtain the whole integral orbital or eyeball sections, an alternative is to use adhesive tape-aided cryo-sectioning. This technique has been well established and successfully applied in histology-sectioning [41]. However, the tape-aided sections can sometimes result in undesirable artifacts such as tears or folds in the steps of staining and dehydrating or clearing, when the adhesive contacts with oil-soluble chemicals and binding forces between the tapes and sections weaken.

In order to circumvent the contacts of the tape section with tape adhesion-diminished oil-soluble reagents during histological processing in the routine staining protocol, we did the orbital or eyeball tissue block staining beforehand, followed by adhesive tape aided sectioning. In the end, the tape-sections were coverslipped with a water-soluble mounting medium, which allows for securing integral and undamaged staining sections. This method was confirmed to be superior compared to 7 other related existing traditional or non-traditional protocols and allows for easy collection of qualified sections with a hundred percent success rate. A trial test on immunostaining with the protocol confirmed the feasibility of this method in detecting target molecules in orbits. Application of the method in other challenging sectioning tissues (e.g., joints, brains, and so on) also produced satisfactory results.

## 2. Materials and methods

### Animal preparation, sample collection, and decalcification

All animal studies (Government of Upper Bavaria; no. 55.2-1-54-2531-25-12), experiments, and procedures were reviewed and approved by the Oberbayern Animal Welfare Committee in Munich, Germany. We confirm that all experiments were performed in accordance with relevant guidelines and regulations.

Female BALB/c mice, with a weight of 18.8–26.4 grams, were delivered from Charles River Laboratory (Sulzfeld, Germany) and allowed to adapt for 1 week before the start of the experiments at the age of 40 weeks. The mice were kept under standard housing conditions.

For orbital preparations, mice were anesthetized and euthanized with isoflurane (CP-Pharma) at a concentration of 1.5–2%. Complete dissection of the orbital and periorbital areas was performed, as described previously [42]. In brief, the heads of the animals were dissected and the skin, the connective tissue around, the brain, and the teeth were removed but left all orbital tissues, eyelids, and adjacent tissues intact. Trimmings were made with coronary cuttings at the positions of Bregma +1.95 and +5.85 mm. Tissue blocks were fixed in 4% paraformaldehyde at 4°C temperature overnight. Decalcification in 15% EDTA (w/v) (ethylenediaminetetraacetic acid) was carried out for 21 days with three times of changing solution, each for one week.

## Tissue block HE (hematoxylin and eosin) staining

After post-fixation in Bouin's (containing 30 mL of saturated picric acid, 10 mL of concentrated formaldehyde, and 2 mL of glacial acetic acid) for 24 h in a hood [buffer formaldehyde (4%, v/v) may serve as an alternate for Bouin's], the tissue block (~8x6x12 mm$^3$) was subjected to tissue block staining with a modified protocol of the literature [43], and all the tissue blocks during the block staining were processed under ultrasound (Type: RK 100, Bandelin Sonorex, Berlin, Germany). High temperature (e.g., > 60°C) due to ultrasound running should be avoided with counteraction of adding some ice to the water sink. Briefly, the tissues went through 2 h running tap water, dehydration with 80% and absolute ethanol, and clearance with xylol, and were then rehydrated with absolute ethanol and water (repeated for 3 times in each step and 40 min for each time). Tissue block HE staining was started with soaking in Harris' hematoxylin solution (Hematoxylin Solution, Harris Modified, HS32, Sigma-Aldrich, Germany) for 18 h and were then rinsed in distilled water and 2% acetic acid prepared in 80% ethanol for 2 hrs each to differentiate the tissue, and the tissues were blued with 1% ammonium water and the tap running water for 2 and 3 h respectively. Finally, Eosin counterstaining was performed with 0.5% (w/v) eosin Y (Cat 3137.2, Carl Roth, Germany) prepared in 15% sucrose (w/v) for 18 h.

## Embedding, sectioning, and mounting

For the preparation of embedding mold, a slip of a rectangular aluminum foil was rolled onto a 12 mm diameter cylinder and fastened with a piece of adhesive tape. Care was taken to adjust the border of one side of the foil aligned or on a level. A foil tube without a bottom was made after the foil was disengaged from the cylinder (see step-by-step protocol of BTA in Supporting Information in S1 File).

For tissue embedding, the tissue blocks were incubated in 3 changes' OCT (Optimal cutting temperature compound, VWR Chemicals, Leuven, Belgium) under ultrasound, each for 1 h, and a slide of glass (1x25x75 mm$^3$, #0656.1, Roth, Germany) was placed on a metal block, which had been frozen in dry ice beforehand. One drop of OCT was put on the slide and the oriented orbital tissue block on the OCT immediately after. Special care was taken to embed the optical nerve approximate end side down. The aluminum mold was sheathed onto the specimen and filled with OCT. The frozen tissue block was ready for sectioning (see step-by-step protocol of BTA in Supporting Information in S1 File).

For the selection of adhesive tape, we preferred Tesa 57405 (Tesa, Beiersdorf, Hamburg, Germany) since the adhesive is powerful in low temperature and adhesive tape (two elements-adhesive and plastic film) is colorless and transparent in itself. More importantly, even, the price is very cheap in comparison with commercially available special adhesive tapes, such as Japanese adhesive film (Cryo-film type IIC9, SECTION-LAB, Japan) and it is easy to obtain.

For the adhesive tape-aided sectioning, the tissue block was mounted onto the cryostat chuck using OCT after peeling off the foil mold. The cryostat (CM1850 cryostat; Leica Biosystems, Buffalo Grove, IL) was set at 5 μm section thickness, 5°cutting plane angle, and -20°C chamber temperature. The tissue block was balanced in the chamber for at least 45 min before starting sectioning. A slip of the adhesive tape (15 × 20 mm) was held with fine forceps at a corner of handle, and pressed with the adhesive side of the tape onto the trimmed surface of the sample block. For best adhesion results, light pressure was applied on the tape with soft tissue. It was ensured that the sample cutting surface was entirely covered by the tape, and the cutting blade was sharp. The sample was sectioned slowly, evenly, and continuously without the use of a brush or anti-roll device. A pair of forceps was used to pick up the freshly sectioned tissue by

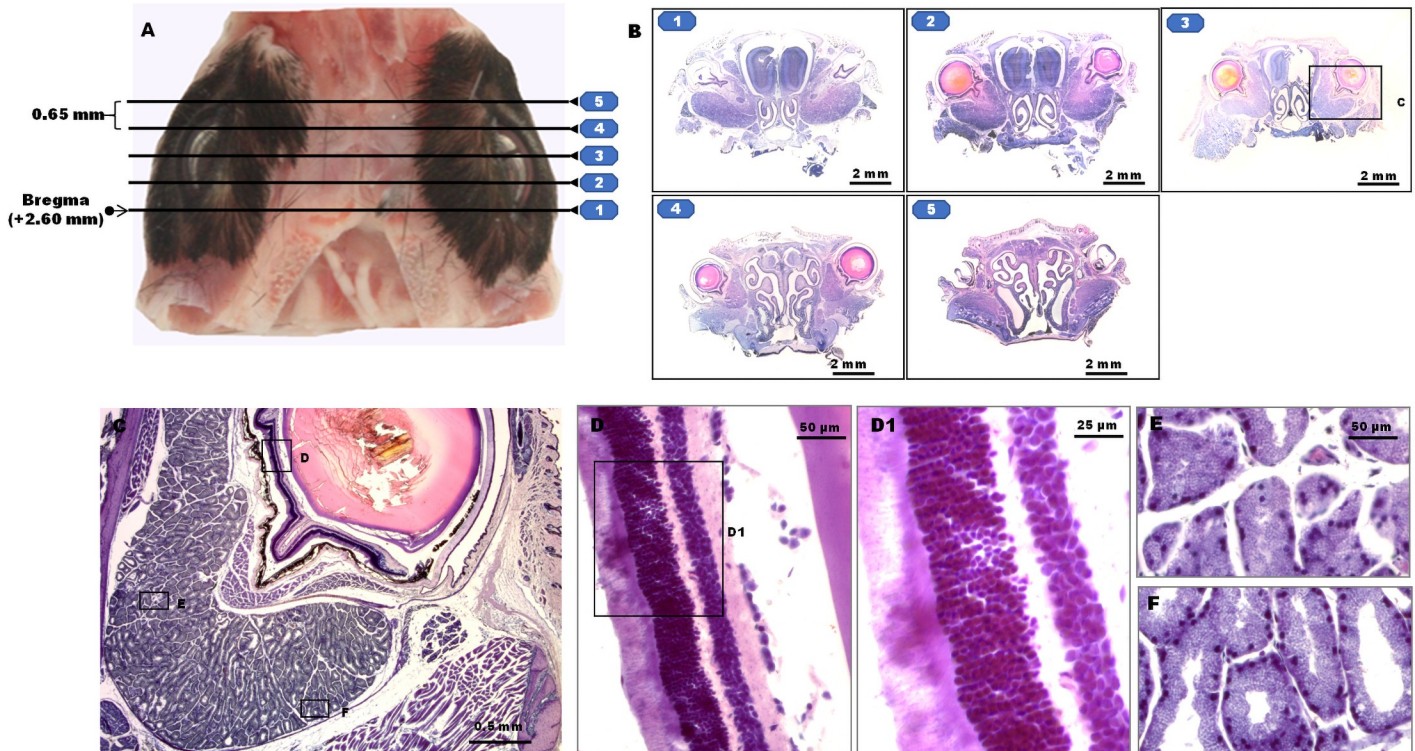

**Fig 1. Images of the orbit or eyeball from a 40 weeks'old BALB/c mouse.** The specimen is processed with the protocol of BTA (block tissue staining, tape-aided sectioning, and aqueous mounting). Serial coronary sections of 5 μm thickness taken at distances of 650 μm are cut at the positions shown in (A). Images from each cutting position are shown in (B). An image of the right orbit magnified in the middle of the serial (Bregma, +3.90 mm) is shown in (C). Images (D-F) are local magnifications of (C) and D1 is the local magnification of (D).

the handle of the tape. The section was then dipped into 20% glycerol two times, placed onto a piece glass of slide with section side up, and finally coverslipped.

Using the microtome, we performed serial coronary sections (5 μm thick, 0.65 mm apart). Cutting was started at the level of Bregma, +1.95 mm, and sections were started to be collected at positions, +2.60, +3.25, +3.90, +4.55, and +5.20 mm (shown in Fig 1) as we did before [42]. The collected sections represented a total distance of 3 to 4 mm and covered the whole eyeball and orbital region of each mouse head.

## Imaging and acquisition

The macro examination was carried out on the orbital staining sections under an epi-microscope (LED-lupenleuchten, Cat NH99.1, Carl Roth, Germany) and the image was captured with a Canon digital camera (Canon EOS 600D) and recorded with 3456 x2304 pixel resolution.

The HE staining resultant sections were viewed under field illumination and the immune fluorescent staining sections were examined using a filter set (550 nm excitation and 570 nm emission) for Cy-3 and a filter set (365 nm excitation and 450 nm emission) for DAPI, on a Zeiss upright microscope (Carl Zeiss AG, Oberkochen, Germany) using a 2.5X, 10X, and 20X or 40X objective lens (Axioscope, Carl Zeiss). The photos were acquired with an Axiovision digital camera system and recorded with 2560x1920 pixel resolution each.

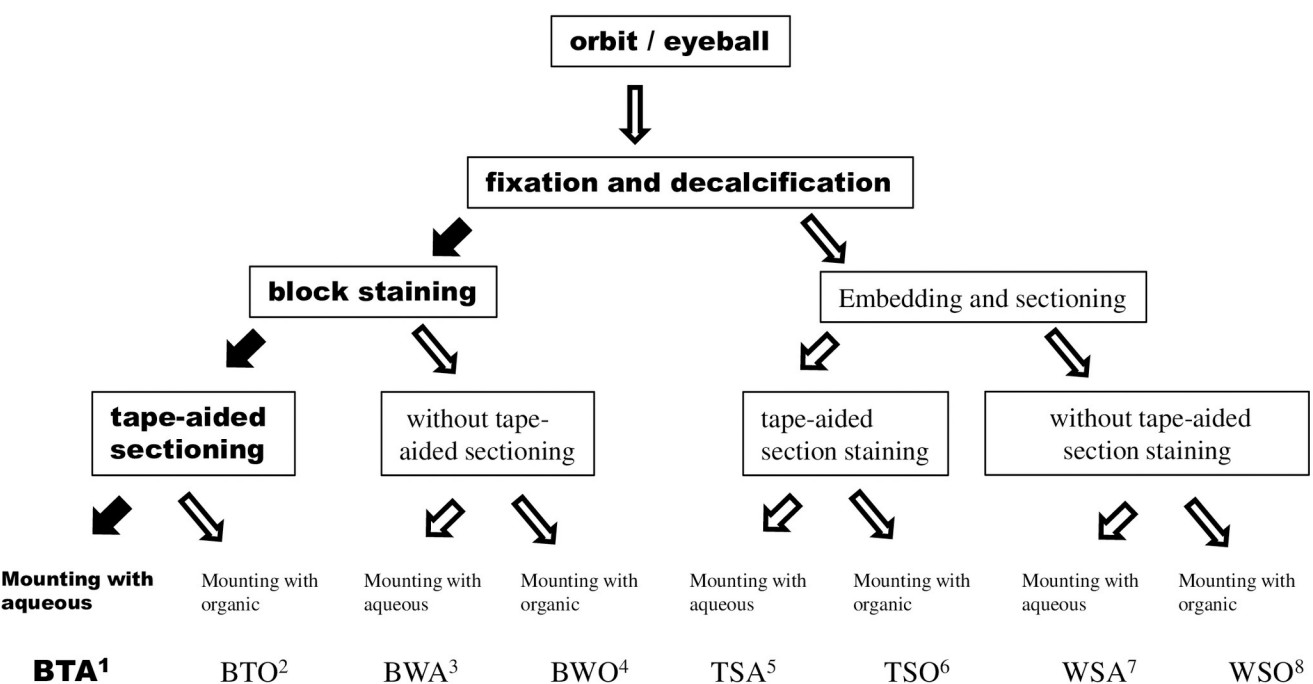

1. **BTA**, B̲lock staining, t̲ape aided sectioning, and a̲queously mounting.
2. BTO, B̲lock staining, t̲ape aided sectioning, and o̲rganically mounting.
3. BWA, B̲lock staining, w̲ithout tape aided sectioning, and a̲queously mounting.
4. BWO, B̲lock staining, w̲ithout tape aided sectioning, and o̲rganically mounting.
5. TSA, T̲ape aided sectioning, s̲taining, and a̲queously mounting.
6. TSO, T̲ape aided sectioning, s̲taining, and o̲rganically mounting.
7. WSA, w̲ithout tape aided sectioning, s̲taining, and a̲queously mounting.
8. WSO, w̲ithout tape aided sectioning, s̲taining, and o̲rganically mounting.

**Fig 2. Flow chart of different protocols.**

## Comparisons of 8 related protocols

For comparisons with other 7 related existing protocols, 24 mouse heads (a pair of orbits or eyeballs for each head) were employed and processed histologically as shown in Fig 2. Note that the experimental protocol, which was described above in detail is highlighted in bold (see Fig 2).

12 randomly selected heads were subjected to tissue block staining, followed by tape-aided sectioning in 6 heads and without tape-aided sectioning in 6 remaining heads, and then cover-slipping with either water-soluble (20% glycerol in water) or oil-soluble mounting medium DPX (synthetic mounting medium, Sigma) in 3 heads each (refer to Fig 2). The protocol of tissue b̲lock staining, t̲ape-aided sectioning, and a̲queous mounting was termed as BTA; the protocol of b̲lock tissue staining, t̲ape aided sectioning, and o̲il-soluble mounting as BTO; the protocol of b̲lock tissue staining, w̲ithout tape aided sectioning, and a̲queous mounting as BWA; the protocol of b̲lock tissue staining, w̲ithout tape aided sectioning, and o̲il-soluble mounting as BWO.

The remaining 12 heads were subjected to traditional histological processing,—sectioning with or without adhesive tape aided after tissue block embedding, conventional HE staining of the sections, and finally mounting (refer to Fig 2). The protocol of tape-aided sectioning, staining, and aqueous mounting is termed as TSA; the protocol of tape-aided sectioning, staining, and oil-soluble mounting as TSO; the protocol of without tape-aided sectioning, staining, and aqueous mounting as WSA; the protocol of without tape-aided sectioning, staining, and oil-soluble mounting as WSO.

Each protocol was applied in 3 mouse heads. For comparisons of qualified section yield rates and slice qualities among different related protocols, 10 serial sections were cut at the position of Bregma, +3.90 mm in each animal head. A total of 30 sections were produced for each group of one method. A total of 240 sections (8 methods in total) was subjected to the corresponding treatments mentioned above (Fig 2). The final high-quality staining sections were counted after coverslipping for the determination of the qualified section yield rate. One best quality section of the right-side orbital tissues among the ten for each head was selected for slice quality comparisons among the various protocols according to the criteria of staining section quality.

## Determination of qualified section success rate

Some sections were lost during processing. For the unified standard, we took each of the following three facts as one section loss. They included the failure of section mounting, section falling off during processing, or more than half of stained section area damaged or covered with a section fold. The remaining sections (exclusive of section loss) were counted and the qualified section yield rate was defined as the ratio between the quantity of the remaining sections and total sectioning sections (referred to as 30 sections) in each protocol.

## Staining section quality evaluation

The total or highest score was 12 for the best quality staining section without any flaws found. A staining section quality score for each section was determined by a deduction of 1 at every one artifact observed from 12. These items of artifacts are listed in Table 1 on three levels of magnification. All sections were evaluated in a blinded fashion (ZL). The average of the scores of the three sections for each protocol stands for the histological quality score of the method.

**Table 1. Artifacts observed on a staining section.**

| Magnification (objective) | Flaws |
|---|---|
| **Macroscopic** | **section wrinkle or fold** |
| | **bubble under section** |
| | **tissue fissure** |
| | **missing part of tissue** |
| **2.5 X** | **contaminant(e.g., dye dregs) adhere** |
| | **shattering** |
| | **local tissue dislocation** |
| | **distorted tissue structure** |
| **10 X** | **blurring view** |
| | **poor staining color contrast** |
| | **indistinct tissue structure** |
| | **unusual cell or tissue aggregation** |

## Tissue block immunostaining of orbits

The orbital tissues were prepared as described above. For tissue block immunostaining, a one-step method was used to localize mouse IgG. The decalcified orbits were incubated with a combinational medium [10 wt% of Triton X-100 (#3051.1, Roth, Germany), 5 wt% of Tween-20 (#9127.2, Roth, Germany), 0.3 wt% of Albumin V (#0052.1, Roth, Germany), 50 mM of Glycine (#3908.3, Roth, Germany), and 0.05% of NaN3 in 0.2 M PBS] and shaken for 48 h at room temperature, in combination to delipide and block unspecific bindings. Then, the samples were washed in PBS of shaking for 24 h and stained with Cy3-conjugated Goat anti-mouse IgG (#115-165-062, Jackson ImmunoResearch Lab. Inc.) at a concentration of 15 μg/ml for 5 days at room temperature. The antibody was prepared in the diluted combinational medium with 0.1 M PBS (1:1). After staining, these samples were immersed and washed in 30% sucrose in PBS and subjected to embedding, and cryosectioning. As a negative control, no antibodies were applied. For the sections in which fluorochrome-conjugated antibodies were applied or those as the corresponding negative control, the mounting medium containing DAPI (#H-1200, Vector Labs) was used for coverslipping. As a control of a traditional way (e.g., WSA, without tape aided sectioning, staining, and aqueous mounting) of immunostaining, the sections were prepared after fixation, decalcification, embedding, and sectioning. After post-fixation with acetone for 10 min and unspecific blocking with a blocking medium [1% (w/v) of Albumin V (#0052.1, Roth, Germany), 25 mM of Glycine (#3908.3, Roth, Germany), and 2% (v/v) of goat serum (DAKO, Hamburg, Germany) in 0.01 M PBS] for 30 min, Cy3-conjugated Goat anti-mouse IgG was applied on the sections and incubated at 4˚C overnight. The antibody was prepared in 0.01 M PBS at a concentration of 3 μg/ml.

The coronary sections used were collected at the level of Bregma, +3.90 (refer to Fig 3), and the regions indicated with (C) (in the image B of Fig 3) were viewed for comparison.

## Data analysis

Data are presented as means ± SEM. SPSS (IBM SPSS Statistics for Windows, Version 11.0., IBM Corp., USA) was employed for multiple comparisons of means. A $p < 0.05$ was considered statistically significant.

## 3. Results

### Comparative evaluation of 8 methods

Tissue block staining and domestic adhesive tapes (e.g., BTA) produced qualified integral serial cryo-sections of whole orbits or eyeballs in adult mice, and the fine structures were well preserved (Fig 1). This method proved to be the best in comparison with seven other related existing histological traditional or non-traditional processing methods, according to the staining slice quality (Fig 4, for more detail, refer to S10 Table in Supporting Information in S1 File). Representative images processed by 8 techniques are shown in Fig 5. All the slice quality scores were compared using one-way ANOVA and Turkey post-test for multiple comparisons of SPSS, resulting in significant differences. A hundred percent success rate in sectioning, collection, and mounting with this method of BTA, is reflected in Table 2.

### Reproducibility of BTA

A reliable method should be reproducible and not be affected by day-to-day affected variation. The reproducibility tested on qualified section success rates and slice quality scores with BTA (see the protocol in the Supporting Information in S1 File) were carried out in four different periods of time. To test the repeatability of the procedure of BTA, 8 heads of male DBA1/J

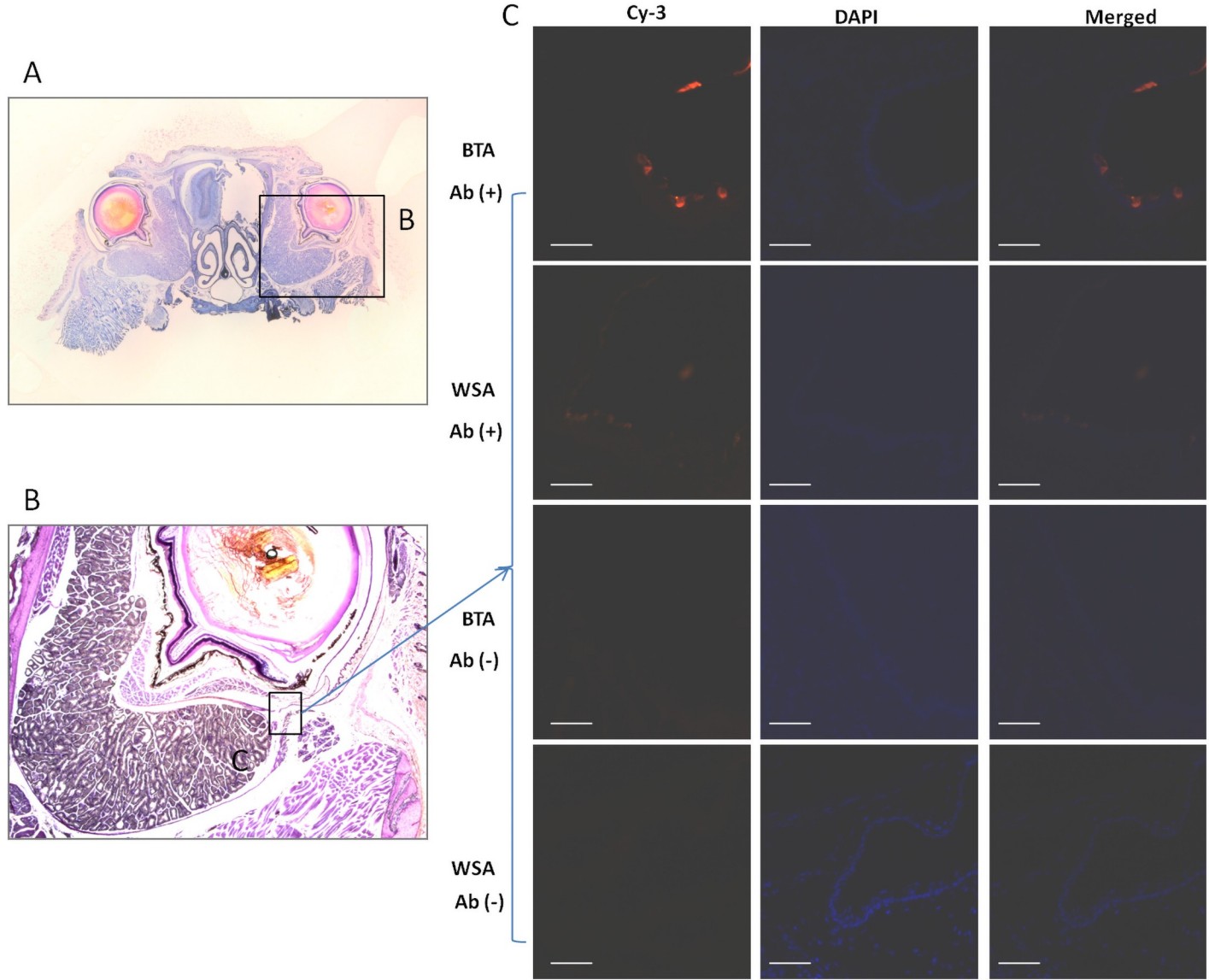

**Fig 3. Images of immunostaining for IgG in orbits of native mice.** The sections of (A) are collected at the position of Bregma, +3.90 mm. B. The right orbital image magnified in the local of (A) displays a region (inset) of (C), in which the fluorescent images in the right panels (C) were taken. Bars indicate 50 μm.

mice (Janvier Labs, France), 33 weeks in age, were objected to the same processing as described above. The evaluations of slice quality scores and qualified section success rates (ZL) were completed in four separate experiments in different periods of time, with two heads for each. The results, as shown in Table 3, were in high similarity among four runs of measurements, which indicates a high reproducibility.

## Application of BTA in immunostaining of orbits

In the eye research field, immunostaining of mouse orbits for cell-specific and molecular markers is critical. In order to extend the method's potential application to immunostaining, we tried tissue block immunostaining in a simple and familiar way for a start. We selected the

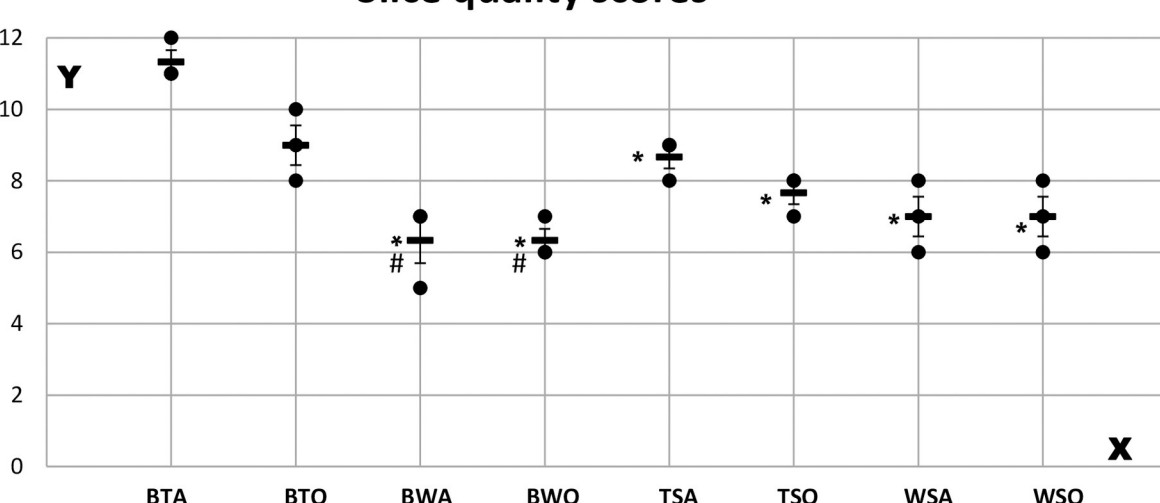

**Slice quality scores**

One-way multiple comparisons among the different protocols, with Tukey HSD of ANOVA result in statistical significance.  N=24,  *, vs BTA, p=0.0225 − 0.000041; #, vs BTO, p=0.0225.

**Fig 4.  Slice quality comparison among different protocols.** The average slice quality scores of the 3 best staining sections at the position of Bregma +3.90 mm (the best quality section among 10 in one head of animal) in the 8 protocols (X-axis) are shown in Y-axis.

target of mouse IgG, which is often used in our laboratory to assess the severity of edema [44], as an example. On the other hand, there are abundant vessels including venous plexus or sinus and arteriolae around an eyeball in the normal state of native mice. IgG is trapped in the blocks of blood coagulation, which often occurs inside the vessels of the dead animals. Immunostaining with BTA for IgG was performed and the results demonstrated the target is present in the vessels (see Fig 3). As a negative control, there were no signals found at the corresponding position. In comparison with the immunostaining in the traditional way (e.g., WSA), a stronger signal was found in the image by BTA (see Fig 3). The preliminary outcomes displayed a stronger signal staining with BTA, which is at least comparable to the traditional immunostaining with WAS. Overall, we can conclude that immunostaining with BTA for molecular targets is feasible.

It should be stated that we cut the orbital tissues at a thickness of 7μm since it was difficult to obtain a complete 5μm-thick highly qualified section without aid of adhesive tapes during the preparation of the control (WSA). The higher resolution of the image (e.g., objective lens of > 20x) was limited due to the section thickness (7 μm).

### Application of the method (BTA) in other technically sectioning-challenged tissues

Application of BTA in joints, brains, and so on,—the challenging integral sectioning tissues, also generated high-quality histological staining sections (refer to S6-S9 Figs in Supporting Information in S1 File).

### 4. Discussions

Generating high-quality histological sections of animal tissues can be technically challenging when using traditional staining protocols that preserve tissue integrity of morphology. These

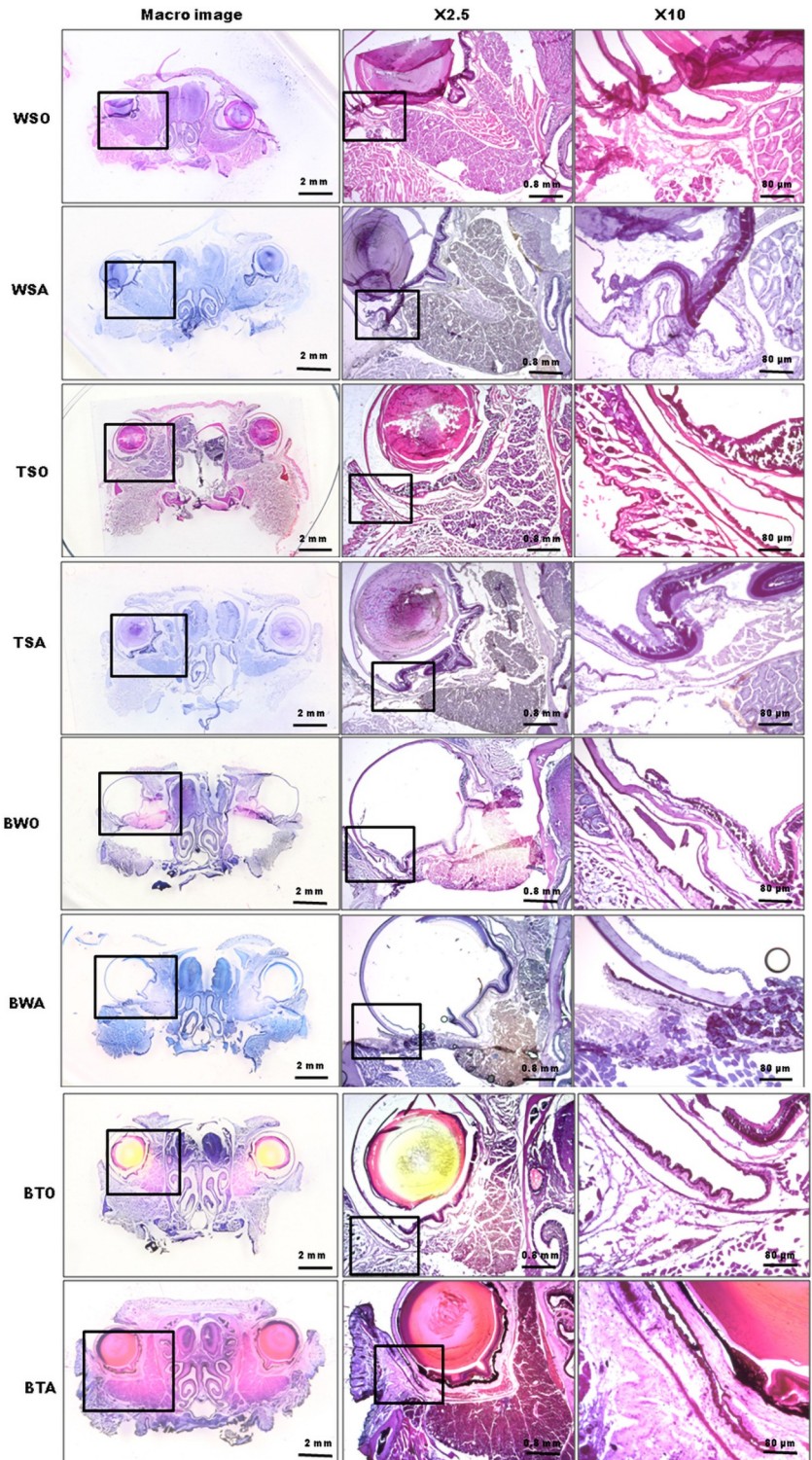

**Fig 5. List of representative images of 8 protocols.** The representative best quality orbital and eyeball staining images for each protocol are shown at three objective magnification levels—macro (gross), X2.5, and X10. Images in Panel X10 and Panel X2.5 are the local magnifications in those of Panel X2.5 and Panel Macro respectively.

**Table 2. Qualified staining section success rate.**

| BTA (%) | BTO(%) | BWA(%) | BWO(%) | TSA(%) | TSO(%) | WSA(%) | WSO(%) |
|---|---|---|---|---|---|---|---|
| 100.0 (30) | 93.3 (28) | 80.0 (24) | 76.7 (23) | 83.3 (25) | 80.0 (24) | 73.3(22) | 66.7(20) |

Figures in the parentheses indicate the values of the qualified staining sections in 30 collected.

challenges are especially obvious when cutting thin sections (less than 10 μm in thickness) of whole orbits or eyeballs from adult animals. Even with every effort in sectioning, the sections displayed curling, distortion, and missing, as seen in traditional staining, e.g., WSO of Figs 4 & 5, and remained so although the lens had been removed prior to sectioning in some cases.

These phenomena are due to markedly various shrinkage of different tissues under different temperatures [34]. As soon as contact between the warm slide and the frozen section during cryo-sectioning is made, the section will immediately melt, which will then convert the ice crystals back into the water. The flow of the melted solutions can distort and displace fine structures because of different shrinkage. At this point, the melted section is dried and surface tension forces continually distort, displace, and collapse tissue structure. For example, the retina detached from the choroid or a huge gap between retina and choroid was often observed after sectioning. Even by chance, an intact section of orbital or eyeball is secured, but it becomes incomplete or displaces after conventional staining. The same phenomenon was observed during section flattening and collection when the paraffin sections float on the warm water. Thus, histological preparation of only parts of the orbital or eyeball has become a way to overcome at least some of the problems.

To preserve the integral and fine tissue structures, an alternative is to use adhesive tape or film. The adhesive on the sticky tape behaves both like a solid and a liquid. Like a liquid, it flows finding its way into any tiny gaps in the surface it is being stuck to when a little pressure is applied on it. It turns back into a solid when the pressure is withdrawn, allowing it to lock into those gaps and to hold itself in place [45]. Thus, the adhesive tape would help to stabilize, support, and capture sections.

Therefore, we used an adhesive tape-aided sectioning technique and traditional staining protocols in the orbital or eyeball tissue. In this procedure, adhesive tape was attached to cutting tissue surfaces, and when the block was sectioned and detached, the cut section remains adhered to the tape without curling or other deformation. As seen in TSO of Figs 4 & 5, however, although the staining section quality was much better than those of WSO, undesirable artifacts were still found. These artifacts included displacement and wrinkles of collagen in the interior wall of the orbital.

Contacts of the section-tape with these oil-soluble reagents such as ethanol or xylol during staining, dehydrating, clearing, and mounting was supposed to contribute to weakened bonding forces between section and adhesive tape [46], which will lead to distorting the intact

**Table 3. Repeatability tests on slice quality scores and qualified section success rates with BTA in 4 separate experiments.**

|  | First ran | Second ran | Third ran | Fourth ran |
|---|---|---|---|---|
| Success rate (%) | 90 (18) | 100 (20) | 100 (20) | 90 (18) |
| Quality scores (means±SEM) | 10±2.02 | 12±2.02 | 11±2.02 | 10±2.02 |

The high qualified sections and success yield rates are obtained from these 4 different runs. 20 sections are collected each round. Qualified sections are counted as indicated in parentheses.

sections as seen in TSO in Figs 4 & 5. Staining secondary to tape-aided sectioning as in TSO means that the tape—sections are contacted with oil-soluble solutions.

To circumvent the contact between the sections and adhesion diminished oil-soluble reagents, we developed a method of using a tissue block staining before the adhesive tape-aided sectioning, staining and dehydrating and clearing and oil-soluble mounting as the traditional staining procedures, allowing for intact cryo-staining-sectioning and high-quality staining sections as seen in Figs 1, 4 and 5 and S10 Table in S1 File.

According to the staining section quality score, the protocol of BTA is best in comparison with 7 other related existing protocols (refer to Fig 4). These 7 methods included either traditional methods of sectioning (with or without tape aid)–staining—mounting or nontraditional ones of block tissue staining—sectioning (without tape aid)—mounting or block tissue stained —tape-aided sectioning but mounting with oil-soluble medium (for more detail, see Figs 2 & 5).

Besides, with the protocol of BTA, the number of sections lost was greatly reduced after final coverslipping. 30 serial sections collected after sectioning yielded 30 intact and high-quality staining sections (refer to Table 2). The staining sections displayed high quality, free of folds or tears, and the faithful preservation of the fine structures (see Fig 1). The structure orientation was automatically achieved. This feature becomes important in applications requiring exact alignment of consecutive sections, such as comparisons among the different treatment groups and researches on the specific local site modification of retina in visual neuroscience It is most applicable to large, high-quality, and thin (<10 μm) serial sections, such as for 3-D digital reconstruction.

The repeat measurements on section quality scores and qualified section success rates for BTA protocol, as shown in Table 3, were in high similarity among the four runs. The high reproducible data confirmed that the technique is reliable over time.

Application with this protocol in the challenging integral sectioning tissues, such as joints (S6 Fig in S1 File), brain (S7 Fig in S1 File), heart or kidney (S8 Fig in S1 File), and lung or spleen (S9 Fig in S1 File) also generated high-quality histological sections. The cell structure with the protocol of BTA was well preserved without significant distortion.

The method's potential application in immunostaining to detect molecular markers was also successfully implemented in our preliminary experiments. Its successful application in immunostaining will benefit somewhat in eye research.

In addition, HE is the most commonly used stain for light microscopy in histopathology laboratories due to its comparative simplicity and ability to demonstrate a wide range of both normal and abnormal cell and tissue components [47]. Whole intact sections and good quality images at relatively high magnification are essential for digital pathology [48]. Therefore, this method is also very suitable for analyzing whole slide images in digital pathology and thus able to provide global information for quantitative and qualitative image analysis. Digital pathology is a rapidly growing field, offering such advantages as remote diagnostics and the application of image analysis to improve the efficiency of the decision process [48–50].

## Turnaround speed, section thickness, and tissue block size

The protocol of BTA originally designed to prepare orbital or eyeball staining sections, is also excellent for the preparation of other tissue staining sections. Using the protocol of BTA, it normally takes 3 or 4 days (exclusive of 1 day for tissue fixation and 21 days for decalcification) to acquire qualified staining sections from a tissue block. Sections as thin as 5 μm or above can be reliably prepared for the present setup. The largest specimens cut in the present study have been for coronal sections of an adult mouse brain, which is encompassed by a 15 x 25 x10 mm$^3$ block. The block size can be increased considerably more than this.

## Limitations

It should be stated that this technique (BTA) was found unsuitable for the liver, in which hematoxylin penetrated two millimeters in-depth (5 mm in total thickness) within 18 hrs of staining. Prolongation of incubation in hematoxylin solution or thinner thickness of the tissue block will be necessary for the liver. In addition, a single staining method (e.g., HE) with this technique is sometimes a drawback if a tissue requires special staining besides HE. But in general, HE is sufficient for the examination of eye histology [33]. Like other aqueous mounting ways, the staining sections with this method should be examined and photographed in 3 weeks, otherwise, they will be dry over 4 weeks. Should this occur, the staining sections can be retrieved by impregnation of one or two drops of aqueous mounting media on the border of the coverslip. Uneven HE staining, which sometimes occurs, may prevent the use of these sections for morphometric studies. But if the appropriately staining and decoloration processes are strictly controlled, the phenomenon can be avoided.

In summary, we have developed a robust cryo-histological method that allows imaging of the whole orbital or eyeball staining sections while maintaining excellent cellular and subcellular morphology. Its applications are not limited to orbit or eyeball, brain, and joints, and to HE staining.

## Supporting information

**S1 File. Supporting information.** Complementary information of step-by-step protocol of BTA (Block staining, tape aided sectioning, and aqueously mounting), applications of BTA in the tissues rather than orbits, and original observational data.
(DOCX)

## Acknowledgments

We acknowledge the excellent technical assistance of Isabel Fodor.

## Author Contributions

**Formal analysis:** Zhongmin Li.

**Investigation:** Zhongmin Li, Julia Faßbender, Clara Wenhart, Hans-Peter Holthoff.

**Methodology:** Zhongmin Li.

**Project administration:** Goetz Muench.

**Resources:** Julia Faßbender, Clara Wenhart.

**Supervision:** Martin Ungerer, Hans-Peter Holthoff, Goetz Muench.

**Writing – original draft:** Zhongmin Li.

**Writing – review & editing:** Zhongmin Li, Martin Ungerer.

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
