## [Decision Letter · Decision Letter 0]

18 May 2021

PONE-D-21-05943

Tissue block staining and domestic adhesive tape yield qualified integral sections of adult mouse orbits and eyeballs

PLOS ONE

Dear Dr. Li,

Thank you for submitting your manuscript to PLOS ONE. I would like to apologize for how long it has taken to provide you with reviews.  After careful consideration, we feel that it has merit but does not fully meet PLOS ONE’s publication criteria as it currently stands. Therefore, we invite you to submit a revised version of the manuscript that addresses the points raised during the review process.

The major concerns of the reviewers concerned your statistical analysis and the quality of the images:  You should use ANOVA followed by a post hoc test for multiple comparisons, rather than t tests.  As far as the images are concerned, the reviewers may have been interpreting the pdf versions, rather than the high definition tiff images that can be downloaded from the website.  However, reviewer 2 points out that fluorescence microscopy and immunofluorescence (IF) staining of rodent orbits with cell-specific and molecular markers are critical. The authors should assess the morphological outcomes of immunofluorescence staining with this method,

We look forward to receiving your revised manuscript.

Kind regards,

Alfred S Lewin, Ph.D.

Academic Editor

PLOS ONE

Journal Requirements:

2. To comply with PLOS ONE submissions requirements, please provide methods of sacrifice in the Methods section of your manuscript.

4. Thank you for stating the following in the Financial Disclosure section:

[The author(s) received no specific funding for this work.].   

We note that one or more of the authors are employed by a commercial company: Advancecor GmbH

Reviewers' comments:

Reviewer's Responses to Questions

**Comments to the Author**

1. Is the manuscript technically sound, and do the data support the conclusions?

Reviewer #1: Yes

Reviewer #2: Partly

2. Has the statistical analysis been performed appropriately and rigorously? 

Reviewer #1: No

Reviewer #2: Yes

3. Have the authors made all data underlying the findings in their manuscript fully available?

Reviewer #1: No

Reviewer #2: No

4. Is the manuscript presented in an intelligible fashion and written in standard English?

Reviewer #1: Yes

Reviewer #2: No

5. Review Comments to the Author

Reviewer #1: Li and colleagues present a study comparing eight cryo-histological protocols for both quality and repeatability. The article is well written, the methods are clearly described, and the data will be beneficial for the scientific community. Though the paper is directed towards identifying the BTA method as both the most reproducible and highest quality, the authors present images of tissue from all eight methods in the supplemental data. Perhaps this figure could be moved to the main part of the manuscript, as it is central to the hypothesis tested in this study. These qualitative impressions would pair nicely with the quantitative data in Figure 3. My only concern of any significance is that it would appear that a student-t test was done in which the score from each technique was compared to the BTA technique. If the experimental design was to prepare tissue with eight different methods, evaluate the sections and identify which method produced the best result, a student-t test would not be the correct statistical tool. Perhaps a one-way ANOVA with a post-hoc Tukey-Kramer multiple comparisons test would be more appropriate?

Specific Comments:

Figure 1, 4, 5: The image quality in Figure 3 was acceptable. The image quality in Figure 1 is so poor I cannot evaluate the tissue. The image quality in 4, 5, and 6 is better than Figure 1, but still not great. Please increase the resolution. The images in the Supplemental Materials were of high quality.

Figure 2: This is extremely helpful and a wonderful visual aid towards providing the reader with a visual explanation of the experimental design.

Figure 3A: BTA – would the scale bar appear more clear in black? Perhaps this is why it not aligned with the other scale bars.

Figure 3B: Please label the y-axis. If 12 was the largest value (perfect score), why does the graph go to 14?

Supplemental Table 3: Are the quality scores presented an average? If so, please provide the SEM.

The data points behind the means were not provided in this study, which I believe is required (Figure 3/Table2/Table S3).

Additional comments:

Line 68 there is an extra space between “easy” and “collection”

Reviewer #2: The manuscript entitled “Tissue block staining and domestic adhesive tape yield qualified integral sections of adult mouse orbits and eyeballs” describes a new frozen sectioning method adapted to keep the morphology of rodent orbit morphology intact. The authors systematically evaluate the outcomes of routine H&E-stained sections with this new method compared with those obtained with other histology methods.

Major concerns relate to insufficient validation of the proposed approach.

1. In the eye research field, immunohistochemical staining techniques with cell-specific and molecular markers are critical. The authors should assess the morphological outcomes of immunoperoxidase stained with this new method, and compare TSA (Tape aided sectioning, staining, and aqueous mounting medium) and WSA (without tape-aided sectioning, staining, and aqueous mounting medium).

2. In the eye research field, fluorescence microscopy and immunofluorescence (IF) staining of rodent orbits with cell-specific and molecular markers are critical. The authors should assess the morphological outcomes of immunofluorescence staining with this new method, and compare TSA (Tape aided sectioning, staining, and aqueous mounting medium) and WSA (without tape aided sectioning, staining, and aqueous mounting medium). If IF is affected, the results should be disclosed with alternate techniques.

3. The authors should determine whether the use of 15% EDTA for 21 days for decalcification of the whole head specimens affects the immunostains. The use optimizes the concentration and duration of immersion with EDTA for immunostaining.

4. Lines 8 and Line 42, "OCT (optimal compound of tissue)" should be "OCT (optimal cutting temperature) compound".

5. Line 213, To compare the quality of section staining, only the best 1 out of the 10 serial sections at the position Bregma+3.90mm was selected for evaluation. They should perform the analysis of the sections at 5 different positions (Bregma +2.60, +3.25, +3.90, +4.55-, and +5.20-mm) as the serial quality of the sectioning is critical for 3D reconstruction.

6. The optical resolution of all the figures is suboptimal photos with better resolution should be provided.

7. The authors should disclose the rate of section loss and the rate of tissue folding. Table 2 showed the success rate for each of the 8 groups. If more than half of the stained section area is damaged or covered by folded tissue, it should be counted as section loss.

8. The authors should add a video as supplementary information.

9. Line 38. The authors should change “…serial whole integrated sections

of orbits or eyeballs to determine further precise site-specific morphological changes that respond to treatments or genetic modifications.” to “…serial whole sections

of orbits or eyeballs to determine morphological changes that respond to treatments or genetic modifications.”

10. Line 43. The tissue thickness should be disclosed.

11. Line 113. The authors should disclose the 3 dimensions of the tissue blocks.

12. Line 117: High concentration of saturated picric acid represents a biosafety risk. An alternate solution should be provided.

13. Line 140, The dimensions of the slide of glass and supplier should be disclosed.

14. Line 170 The authors should write the full name of BTA here.

15. Line 213. The authors state that “One best quality section of the right-side orbital tissues among the ten for each head was selected for section quality comparisons among the various protocols according to staining section quality.” The authors should justify, and explain whether the selection process was performed in a masked fashion.

16. They should indicate whether the person who performed the sectioning was different or not.

17. Line 228: They should indicate the person(s) who assessed the sections with initials.

18. The authors should disclose freezing artifacts such as vacuolization at high power in soft tissue such as retina that can be noticed at high power (X 40 objective). They should also disclose the distribution of freezing artifacts. The authors should show high power images of the deeper structures such as the retina (X 40 objective).

19. The results section should be separated from the discussion section.

20. The results with tissue types other than eye and orbit tissues should be excluded from this manuscript, as the results are preliminary and the quality of the sections has not been validated.

21. Line 367: The authors should disclose the total duration of processing from sacrifice to sectioning.

22. In the limitations section the authors should include uneven HE staining that may prevent the use of these sections for morphometric studies.

6. PLOS authors have the option to publish the peer review history of their article (what does this mean?). If published, this will include your full peer review and any attached files.

Reviewer #1: No

Reviewer #2: No

---

## [Author Response · Author response to Decision Letter 0]

24 Jun 2021

Reviewer #1 

 COMMENT 1:

Li and colleagues present a study comparing eight cryo-histological protocols for both quality and repeatability. The article is well written, the methods are clearly described, and the data will be beneficial for the scientific community. 

RESPONSE:

Thanks for the reviewer’s inspiring comments.

COMMENT 2:

Though the paper is directed towards identifying the BTA method as both the most reproducible and highest quality, the authors present images of tissue from all eight methods in the supplemental data. Perhaps this figure could be moved to the main part of the manuscript, as it is central to the hypothesis tested in this study. These qualitative impressions would pair nicely with the quantitative data in Figure 3. 

RESPONSE:

It is a good suggestion. We have moved the figure to the main part of the manuscript. Please refer to Lines 296 – 303 in blue.

COMMENT 3:

My only concern of any significance is that it would appear that a student-t test was done in which the score from each technique was compared to the BTA technique. If the experimental design was to prepare tissue with eight different methods, evaluate the sections and identify which method produced the best result, a student-t test would not be the correct statistical tool. Perhaps a one-way ANOVA with a post-hoc Tukey-Kramer multiple comparisons test would be more appropriate?

RESPONSE:

We are agreed on reviewer’s points. A one-way ANOVA with a post-hoc Tukey multiple comparisons (unfortunately, there is no such item - Tukey-Kramer found in our SPSS software – V11. Sorry for that) have been made, which replaced the Student T test. The changes have been present in the current manuscript. Please see Lines 277 -280, 291 -294,296 - 298 in blue.

COMMENT 4:

Figure 1, 4, 5: The image quality in Figure 3 was acceptable. The image quality in Figure 1 is so poor I cannot evaluate the tissue. The image quality in 4, 5, and 6 is better than Figure 1, but still not great. Please increase the resolution. The images in the Supplemental Materials were of high quality.

RESPONSE:

The reviewer is correct. The images with a higher resolution – TIFF images are available in the new manuscript. Please refer to TIFF images – Figs 1,3,5 of the revision.

COMMENT 5:

Figure 2: This is extremely helpful and a wonderful visual aid towards providing the reader with a visual explanation of the experimental design.

RESPONSE:

Thanks!

COMMENT 6:

Figure 3A: BTA – would the scale bar appear more clear in black? Perhaps this is why it not aligned with the other scale bars.

RESPONSE:

The viewer is right. We have changed that scale bar with a black one. But we left Fig 3A out in revised manuscript to avoid needless duplication since all the eight resultant representative images have been moved into the main body of the manuscript according to the reviewer’s nice suggestion (Comment 2). 

COMMENT 7:

Figure 3B: Please label the y-axis. If 12 was the largest value (perfect score), why does the graph go to 14?

RESPONSE:

Yes, we have labeled Y- axis and deleted the value of 14 in that Figure. Please note that the figure 3B in earlier version has changed into Figure 4 in the revision due to insertion of new images. Please refer to Lines 296 – 298 in blue.

COMMENT 8:

Supplemental Table 3: Are the quality scores presented an average? If so, please provide the SEM.

RESPONSE:

Yes, the quality scores present average values. We have added SEMs to each corresponding average value. Please see Lines 319 – 320 in blue. 

COMMENT 9:

The data points behind the means were not provided in this study, which I believe is required (Figure 3/Table2/Table S3).

RESPONSE:

The reviewer is right. The complementary data have been added in the corresponding tables or figure of the revised manuscript. Please refer to Fig 4, Tab 2 and Tab 3 in revised manuscript and Lines 296 – 298, 305 -307, 319 – 320 for details in blue. 

COMMENT 10:

Line 68 there is an extra space between “easy” and “collection”

RESPONSE:

Yes, correction has been made in Line 69 of the revised manuscript. 

RESPONSES TO EDITORIAL AND REVIEWER COMMENTS

Reviewer #2 

COMMENT 1:

The manuscript entitled “Tissue block staining and domestic adhesive tape yield qualified integral sections of adult mouse orbits and eyeballs” describes a new frozen sectioning method adapted to keep the morphology of rodent orbit morphology intact. The authors systematically evaluate the outcomes of routine H&E-stained sections with this new method compared with those obtained with other histology methods.

Major concerns relate to insufficient validation of the proposed approach.

1. In the eye research field, immunohistochemical staining techniques with cell-specific and molecular markers are critical. The authors should assess the morphological outcomes of immunoperoxidase stained with this new method, and compare TSA (Tape aided sectioning, staining, and aqueous mounting medium) and WSA (without tape-aided sectioning, staining, and aqueous mounting medium).

2. In the eye research field, fluorescence microscopy and immunofluorescence (IF) staining of rodent orbits with cell-specific and molecular markers are critical. The authors should assess the morphological outcomes of immunofluorescence staining with this new method, and compare TSA (Tape aided sectioning, staining, and aqueous mounting medium) and WSA (without tape aided sectioning, staining, and aqueous mounting medium). If IF is affected, the results should be disclosed with alternate techniques.

RESPONSE:

We are agreed on reviewer’s points. The integral sections of orbits are prerequisite for all the subsequent kinds of staining and morphological evaluation. The first step is to ensure an intact qualified section, which was originally designed for in the method, and the applications of the method are the next steps. 

In order to extend its potential application in immune-staining, as reviewer’s kind suggestions, we made immunostaining in blocks of orbital tissue. That is a new issue, which involves the antibody permeability during 3D immunostaining (Susaki, 2020 NATURE COMMUNICATIONS | https://doi.org/10.1038/s41467-020-15906-5). Delipidation is necessary for the antibody penetration, but it may damage the cell membrane structures (concerning cell-specific markers). Therefore, we selected the target of mouse IgG, which is simple and often used in our laboratory to assess the severity of edema, (Li, 2020 Scientific Report | https://doi.org/10.1038/s41598-020-76950-1) as an example. The modified method from the present one described was adopted for the new application. 

We cut the orbital tissue at a thickness of 7µm since it was difficult to cut a complete qualitified 5µm-thick-section without aid of adhesive tapes in the control (WSA). It is the section thickness (7µm) that an image of a higher resolution (objective lens > 20) was limited. But the images of a lower resolution, which is present in the revision, do not affect our conclusion that BTA application in orbital immunostaing is feasible. For more details, please refer to Lines 188 -189, 243 – 274, 322 -341 in red. 

 We have tried to make immunostaining with HRP-conjugated antibody, but the results were left out in the revised manuscript because of a little too weak signals against background. Here we express our deep regret. 

COMMENT 2:

3. The authors should determine whether the use of 15% EDTA for 21 days for decalcification of the whole head specimens affects the immunostains. The use optimizes the concentration and duration of immersion with EDTA for immunostaining.

RESPONSE:

EDTA (15%) for 21 days or more or less for decalcification is often adopted before sectioning in osteology, odontology and the tissues with calcification. The treatment might affect the antigenicity of the targets for some or maybe not for the others. That is a big issue since there are huge number of different antibodies used for these diversities of the antigens. The different combines of EDTA and fixatives will make the situation more complicated. Our unit is rather small. The limited financial resource cannot afford the investigation at this moment. We would like to express an apology for this.

COMMENT 3:

4. Lines 8 and Line 42, "OCT (optimal compound of tissue)" should be "OCT (optimal cutting temperature) compound".

RESPONSE:

Thanks for the reviewer’s kind reminder. The corrections have been made in the manuscript. Please refer to Lines 8 – 9, 42 in red. 

COMMENT 4:

5. Line 213, To compare the quality of section staining, only the best 1 out of the 10 serial sections at the position Bregma+3.90mm was selected for evaluation. They should perform the analysis of the sections at 5 different positions (Bregma +2.60, +3.25, +3.90, +4.55-, and +5.20-mm) as the serial quality of the sectioning is critical for 3D reconstruction.

RESPONSE:

The reviewer is right. To address the reviewer nicely suggestions, we reassessed the quality of sections at other positions (Bregma +2.60, +3.25, +4.55, and +5.20 mm). The observational outcomes were listed in Supporting Information. Please refer to S10 Table in Supporting Information.

COMMENT 5:

6. The optical resolution of all the figures is suboptimal photos with better resolution should be provided.

RESPONSE:

Yes. The images with a higher resolution – TIFF images are available in the new manuscript. Please refer to TIFF images – Figs 1,3,5 of the revision.

COMMENT 6:

7. The authors should disclose the rate of section loss and the rate of tissue folding. Table 2 showed the success rate for each of the 8 groups. If more than half of the stained section area is damaged or covered by folded tissue, it should be counted as section loss.

RESPONSE:

Yes. We have a new complementary table included in Supporting Information. Please see and also refer to the revised Table 2 (Lines 305 – 307) and S10 Table in Supporting Information.

COMMENT 7:

8. The authors should add a video as supplementary information.

RESPONSE:

We consent to the reviewer’s suggestions. A video will make readers understand the method in a simple and straight-forward way. Unfortunately, our poorly editing technique cannot satisfy the needs of a high resolution and clear images. Instead, we made the presentation, which showed more details with a rather high resolution, to complement the regret. Please see Step-by-Step protocol in Supporting Information.

COMMENT 8:

9. Line 38. The authors should change “…serial whole integrated sections

of orbits or eyeballs to determine further precise site-specific morphological changes that respond to treatments or genetic modifications.” to “…serial whole sections

of orbits or eyeballs to determine morphological changes that respond to treatments or genetic modifications.”

RESPONSE:

OK. The change has been made in the revised manuscript. Please see Line 39 in red. 

COMMENT 9:

10. Line 43. The tissue thickness should be disclosed.

RESPONSE:

Yes. The tissue thickness (the reviewer might mean “ the section thickness”?) should be mentioned there. But we cannot find out the detail value in the author’s book. Sorry we can not offer the value. In general, the value is between 5 – 7 µm. 

COMMENT 10:

11. Line 113. The authors should disclose the 3 dimensions of the tissue blocks.

RESPONSE:

Yes. We inserted the values of the 3D in the new manuscript. Please see Line 121 in red. 

COMMENT 11:

12. Line 117: High concentration of saturated picric acid represents a biosafety risk. An alternate solution should be provided.

RESPONSE:

Yes. We acknowledge a risk using Boun’s fixative, - one saturated picric acid containing solution. But the risk will be kept at a minimum when all the operation related is performed in a hood. Actually, in our trial tests, we have optimized the fixatives. The outcome indicated that the best combination was one between block staining and Boun’s fixative, and suboptimal one is between buffer formaldehyde and block staining. Therefore, an alternate solution should be the fixative of buffer formaldehyde (4%, v/v). Please note that formaldehyde also bears a rather high biosafety risk. Please see Lines 119 – 120 in red.

COMMENT 12:

13. Line 140, The dimensions of the slide of glass and supplier should be disclosed.

RESPONSE:

Yes. We inserted the values of the 3D in the new manuscript. Please see Line 143 in red.

COMMENT 13:

14. Line 170 The authors should write the full name of BTA here.

RESPONSE:

Yes. We have added the full name of BTA there in the new manuscript. Please see Line 176 in red.

COMMENT 14:

15. Line 213. The authors state that “One best quality section of the right-side orbital tissues among the ten for each head was selected for section quality comparisons among the various protocols according to staining section quality.” The authors should justify, and explain whether the selection process was performed in a masked fashion.

RESPONSE:

Yes. The selection process was performed in a masked fashion. Please see Line 240.

COMMENT 15:

16. They should indicate whether the person who performed the sectioning was different or not.

RESPONSE:

Yes. It is the same person who performed the sectioning. But the person is blind to the treatments. What we did facilitates the comparisons in a unified set of criteria. Please see Lines 240,315 in red. 

COMMENT 16:

17. Line 228: They should indicate the person(s) who assessed the sections with initials.

RESPONSE:

Yes. We have included the relative initials in the new manuscript. Please see Lines 240,315 in red.

COMMENT 17:

18. The authors should disclose freezing artifacts such as vacuolization at high power in soft tissue such as retina that can be noticed at high power (X 40 objective). They should also disclose the distribution of freezing artifacts. The authors should show high power images of the deeper structures such as the retina (X 40 objective).

RESPONSE:

We would like to express our regrets. We cannot distinguish the vacuoles due to embedding, freezing, staining, coverslipping and photographing artifacts, or real structures of its. A high resolution image of the retina with the depth (X 40 objective) has been inserted in revised manuscript. Please see D1 of Figure 1. 

COMMENT 18:

19. The results section should be separated from the discussion section.

RESPONSE:

Yes, the separation between Results section and Discussion has been made in the new manuscript. Please see Lines 284,368. 

COMMENT 19:

20. The results with tissue types other than eye and orbit tissues should be excluded from this manuscript, as the results are preliminary and the quality of the sections has not been validated.

RESPONSE:

Yes. We have excluded those parts from the main manuscript and moved to Supporting Information section. 

COMMENT 20:

21. Line 367: The authors should disclose the total duration of processing from sacrifice to sectioning.

RESPONSE:

Yes. We have displayed the span from sacrifice to sectioning in the revised manuscript. Please see Lines 449 – 450 in red.

COMMENT 21:

22. In the limitations section the authors should include uneven HE staining that may prevent the use of these sections for morphometric studies.

RESPONSE:

Yes. We inserted the description in Limitations section of the new manuscript. Please see L466 -468 in red.

---

## [Decision Letter · Decision Letter 1]

15 Jul 2021

Tissue block staining and domestic adhesive tape yield qualified integral sections of adult mouse orbits and eyeballs

PONE-D-21-05943R1

Dear Dr. Li,

We’re pleased to inform you that your manuscript has been judged scientifically suitable for publication and will be formally accepted for publication once it meets all outstanding technical requirements.

Kind regards,

Alfred S Lewin, Ph.D.

Section Editor

PLOS ONE

Additional Editor Comments (optional):

Reviewers' comments:

Reviewer's Responses to Questions

**Comments to the Author**

1. If the authors have adequately addressed your comments raised in a previous round of review and you feel that this manuscript is now acceptable for publication, you may indicate that here to bypass the “Comments to the Author” section, enter your conflict of interest statement in the “Confidential to Editor” section, and submit your "Accept" recommendation.

Reviewer #1: All comments have been addressed

Reviewer #2: All comments have been addressed

2. Is the manuscript technically sound, and do the data support the conclusions?

Reviewer #1: Yes

Reviewer #2: Yes

3. Has the statistical analysis been performed appropriately and rigorously? 

Reviewer #1: Yes

Reviewer #2: Yes

4. Have the authors made all data underlying the findings in their manuscript fully available?

Reviewer #1: Yes

Reviewer #2: Yes

5. Is the manuscript presented in an intelligible fashion and written in standard English?

Reviewer #1: Yes

Reviewer #2: (No Response)

6. Review Comments to the Author

Reviewer #1: I appreciate the authors’ thought comments and responses to reviewer feedback. All significant concerns have been addressed. Of slight note, it appears there is a typo in the last sentence of the Abstract that I missed on the first review: Application of this protocol in joints, brains, and so on, - the challenging integral sectioning tissues, also generated high-quality histological staining sections.

Reviewer #2: The authors addressed all comments and queries.

Aminor suggestion:

Line 273 Bregman should be changed to Bregma.

7. PLOS authors have the option to publish the peer review history of their article (what does this mean?). If published, this will include your full peer review and any attached files.

Reviewer #1: No

Reviewer #2: No

---

## [Editor Report · Acceptance letter]

27 Jul 2021

PONE-D-21-05943R1 

Tissue block staining and domestic adhesive tape yield qualified integral sections of adult mouse orbits and eyeballs 

Dear Dr. Li:

I'm pleased to inform you that your manuscript has been deemed suitable for publication in PLOS ONE. Congratulations! Your manuscript is now with our production department. 

Kind regards, 

on behalf of

Dr. Alfred S Lewin 

Section Editor

PLOS ONE